# Cerebellar transcranial direct current stimulation disrupts neuroplasticity of intracortical motor circuits

**Wei-Yeh Liao, Ryoki Sasaki, John G. Semmler, George M. Opie** *

Discipline of Physiology, School of Biomedicine, The University of Adelaide, Adelaide, Australia

* george.opie@adelaide.edu.au

This is a Registered Report and may have an associated publication; please check the article page on the journal site for any related articles.

## Abstract

While previous research using transcranial magnetic stimulation (TMS) suggest that cerebellum (CB) influences the neuroplastic response of primary motor cortex (M1), the role of different indirect (I) wave inputs in M1 mediating this interaction remains unclear. The aim of this study was therefore to assess how CB influences neuroplasticity of early and late I-wave circuits. 22 young adults (22 ± 2.7 years) participated in 3 sessions in which I-wave periodicity repetitive transcranial magnetic stimulation (iTMS) was applied over M1 during concurrent application of cathodal transcranial direct current stimulation over CB (tDCS$_{CB}$). In each session, iTMS either targeted early I-waves (1.5 ms interval; iTMS$_{1.5}$), late I-waves (4.5 ms interval; iTMS$_{4.5}$), or had no effect (variable interval; iTMS$_{Sham}$). Changes due to the intervention were examined with motor evoked potential (MEP) amplitude using TMS protocols measuring corticospinal excitability (MEP$_{1mV}$) and the strength of CB-M1 connections (CBI). In addition, we indexed I-wave activity using short-interval intracortical facilitation (SICF) and low-intensity single-pulse TMS applied with posterior-anterior (MEP$_{PA}$) and anterior-posterior (MEP$_{AP}$) current directions. Following both active iTMS sessions, there was no change in MEP$_{1mV}$, CBI or SICF (all $P > 0.05$), suggesting that tDCS$_{CB}$ broadly disrupted the excitatory response that is normally seen following iTMS. However, although MEP$_{AP}$ also failed to facilitate after the intervention ($P > 0.05$), MEP$_{PA}$ potentiated following both active iTMS sessions (both $P < 0.05$). This differential response between current directions could indicate a selective effect of CB on AP-sensitive circuits.

## Introduction

The ability to modify patterns of motor behaviour in response to sensory feedback represents a fundamental component of effective motor control. This process underpins our capacity to learn new types of motor skills, and to improve their performance with practice. While this error-based motor adaptation is a complex process involving a distributed brain network, extensive literature has shown that the cerebellum (CB) plays a critical role (for review, see; [1]). This structure is thought to facilitate generation and ongoing modification of internal models of neural activation that determine effector dynamics. These internal models are

**Data Availability Statement:** All deidentified data are made fully available via the open science framework repository (https://osf.io/7et3z/).

**Funding:** The author(s) received no specific funding for this work.

**Competing interests:** The authors have declared that no competing interests exist.

constantly updated based on comparisons between predicted and actual sensory feedback, allowing improved task performance with practice. As an extension of this process, communication between CB and primary motor cortex (M1) is crucial [2,3], and may facilitate retention of the generated internal model [4]. However, the neurophysiological processes underpinning this communication remain unclear, largely due to the difficulty of assessing the associated pathways in human participants.

Despite this, non-invasive brain stimulation techniques (NIBS) such as transcranial magnetic stimulation (TMS) have provided some information on CB-M1 communication. In particular, inhibitory interactions between CB and M1 have been demonstrated using a paradigm called CB-brain inhibition (CBI). This involves applying a TMS pulse over the CB at specific intervals (5–7 ms) prior to a second stimulus over contralateral M1, producing a motor evoked potential (MEP) that is reduced in amplitude relative to an MEP produced by M1 stimulation alone [5–7]. CBI is thought to involve activation of Purkinje cells in CB cortex, leading to inhibition of the dentate nucleus and consequent disfacilitation of M1 via projections through the motor thalamus (for review, see; [8]). Activity of this pathway is known to be modified during the learning of adaptation tasks that rely heavily on input from the CB [9–11], with larger changes in CBI predicting better performance [11].

While this literature demonstrates the capacity of CB to influence M1 in a functionally relevant way, it remains unclear how this influence is mediated. In particular, the circuits within M1 that are targeted by CB are not well understood. Given that previous research using TMS has shown that the activity of specific intracortical motor circuits relates to the acquisition of different motor skills [12], identification of the M1 circuitry that is affected by CB projections may allow the targeted modification of skill acquisition. Interestingly, growing evidence suggests that late indirect (I3) wave inputs on to corticospinal neurons, which represent important predictors of neuroplasticity and motor learning [12–14], may be specifically modified by changes in CB excitability. For example, application of transcranial direct current stimulation (tDCS; a NIBS paradigm that induces neuroplastic changes in brain excitability) over CB specifically modulates paired-pulse TMS measures of late I-wave excitability [15]. In addition, the effects of CB tDCS on single-pulse TMS measures of M1 excitability are only apparent when stimulation is applied with an anterior-posterior current, which specifically activates late I-wave circuits [16]. Also, changes in late I-wave circuits following motor training were observed following a CB-dependent motor task, but were absent following a task with minimal CB involvement [12,17].

Based on this previous literature, it appears likely that CB projections to M1 influence activity within the late I-wave circuitry. However, the nature of this influence, particularly in relation to the plasticity of these circuits, remains unclear. The aim of this exploratory study was therefore to assess how changes in CB activity influence the excitability and plasticity of I-wave generating circuits in M1. To achieve this, I-wave periodicity repetitive TMS (iTMS; [18,19]) was used to induce neuroplastic changes within early (I1) and late (I3) I-wave circuits, while cerebellar tDCS was concurrently applied to modulate the influence that cerebellum has on M1. We reasoned that reducing the inhibitory influence of CB on M1 (via the cerebello-thalamo-cortical pathway) could potentiate the neuroplastic response to iTMS, and that differential responses to iTMS applied with short (targeting early I-waves) and longer (targeting later I-waves) latencies would highlight specific patterns of connectivity between CB and the I-wave circuits. As cerebellar cathodal tDCS has been shown to reduce the inhibitory effects of CB on M1 [20], cathodal stimulation was applied over CB during application of iTMS.

## Methods

### Sample size and participants

While the effects of CB tDCS on iTMS have not been previously investigated, the study by Ates and colleagues [15] investigated the influence of CB tDCS on the excitability of the I-wave generating circuits. Consequently, sample size calculations based on this study were sufficient to demonstrate the effects of activation within the pathway of interest (i.e., cerebellar projections to I-wave circuits of M1). Examination of the findings reported by Ates and colleagues revealed that changes in short-interval intracortical facilitation (SICF; paired-pulse TMS protocol indexing I-wave excitability; [21,22]) due to CB tDCS had an effect size of 0.67. Based on the results of an *a priori* power analysis utilising this effect size, with $\alpha = 0.05$ and $1\text{-}\beta = 0.9$, we recruited 22 individuals (22 ± 2.7 years; 11 female) to participate in the proposed experiment.

All participants were recruited via advertisements placed on notice boards within the University of Adelaide, in addition to on social media platforms. Exclusion criteria included a history of psychiatric or neurological disease, current use of medications that effect the central nervous system, or left handedness. Suitability to receive TMS was assessed using a standard screening questionnaire [23]. The experiment was conducted in accordance with the Declaration of Helsinki, and was approved by the University of Adelaide Human Research Ethics Committee (approval H-2019-252). Written, informed consent was provided prior to participation. All deidentified data are made fully available via the open science framework repository (https://osf.io/7et3z/).

### Experimental arrangement

All participants attended the laboratory for three separate sessions, with a washout period of at least 1 week between sessions. While the experimental protocol applied within each session was the same, the ISI used for iTMS varied between sessions (see below & Fig 1). Furthermore, the order in which each iTMS interval was applied was randomised between participants. As diurnal variations in cortisol could influence the neuroplastic response to TMS [24], all plasticity interventions were applied after 11 am and at approximately the same time of day within each participant.

During each experimental session, participants were seated in a comfortable chair with their hand resting on a table in front of them. Surface electromyography (EMG) was recorded from the first dorsal interosseous (FDI) of the right hand using two Ag-AgCl electrodes arranged in a belly-tendon montage on the skin above the muscle. A third electrode attached above the styloid process of the right ulnar grounded the electrodes. EMG signals were amplified (300x) and filtered (band-pass 20 Hz– 1 kHz) using a CED 1902 signal conditioner (Cambridge Electronic Design, Cambridge, UK) before being digitized at 2 kHz using CED 1401

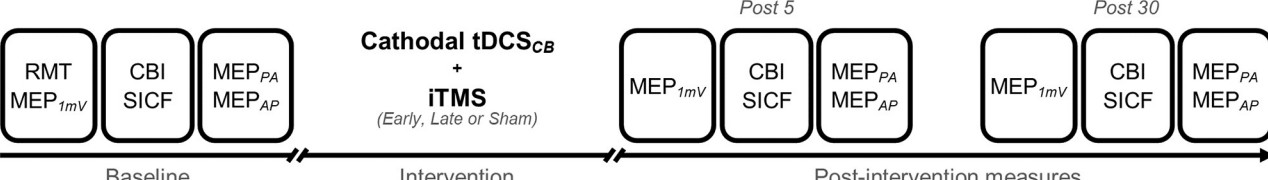

**Fig 1. Experimental protocol.** RMT, resting motor threshold; $MEP_{1mV}$, standard MEP of ~ 1 mV at baseline; CBI, cerebellar-brain inhibition; SICF, short-interval intracortical facilitation; $MEP_{PA}$, standard MEP of ~ 0.5 mV at baseline with PA orientation; $MEP_{AP}$, standard MEP of ~ 0.5 mV at baseline with AP orientation; $tDCS_{CB}$, transcranial direct current stimulation applied to the cerebellum; iTMS, I-wave periodicity repetitive transcranial magnetic stimulation.

analogue-to-digital converter and stored on a PC for offline analysis. Signal noise associated with mains power (within the 50 Hz frequency band) was also removed using a Humbug mains noise eliminator (Quest Scientific, North Vancouver, Canada). To facilitate muscle relaxation when required, real-time EMG signals were displayed on an oscilloscope placed in front of the participant.

## Experimental procedures

**Transcranial Magnetic Stimulation (TMS).** A figure-of-8 coil connected to two Magstim $200^2$ magnetic stimulators (Magstim, Dyfed, UK) via a BiStim unit was used to apply TMS to the left M1. The coil was held tangentially to the scalp, at an angle of 45˚ to the sagittal plane, with the handle pointing backwards and laterally, inducing a posterior-to-anterior (PA) current within the brain. The location producing the largest and most consistent motor evoked potential (MEP) within the relaxed FDI muscle of the right hand was identified and marked on the scalp for reference; this target location was closely monitored throughout the experiment. All pre- and post-intervention TMS was applied at a rate of 0.2 Hz, with a 10% jitter between trials in order to avoid anticipation of the stimulus.

Resting motor threshold (RMT) was defined as the stimulus intensity producing an MEP amplitude $\geq$ 50 µV in at least 5 out of 10 trials during relaxation of the right FDI. RMT was assessed at the beginning of each experimental session and expressed as a percentage of maximum stimulator output (%MSO). Following assessment of RMT, the stimulus intensity producing a standard MEP amplitude of approximately 1 mV ($MEP_{1mV}$), when averaged over 20 trials, was identified. The same intensity was then applied 5 minutes and 30 minutes following the intervention in order to assess changes in corticospinal excitability.

**I-wave excitability.** As assessing the influence of CB modulation on I-wave excitability was the main aim of this project, changes in SICF were the primary outcome measure. This paired-pulse TMS protocol produces MEP facilitation when conditioning and test stimuli are separated by discrete ISIs that correspond to I-wave latencies recorded from the epidural space [21]. SICF utilised a conditioning stimulus set at 90% RMT, a test stimulus set at $MEP_{1mV}$ and two ISIs of 1.5 ($SICF_{1.5}$) and 4.5 ($SICF_{4.5}$) ms, which correspond to the early (I1) and late (I3) MEP peaks apparent in a complete SICF curve [21,25,26]. Measurements of SICF included 12 trials for each condition, at each time point.

As a secondary measure of I-wave function, TMS was applied with different stimulus directions, which altered the interneuronal circuits contributing to the generated MEP [13,27,28]. When TMS is applied with a conventional (PA) current direction, the resulting MEP is thought to arise from preferential activation of early I1-waves. In contrast, when the induced current is directed from anterior-to-posterior (AP; coil handle held 180˚ to the PA orientation), the resulting MEP is thought to arise from preferential activation of later (I2-3) I-waves. The stimulus intensity producing an MEP of approximately 0.5 mV was assessed for both PA ($MEP_{PA}$) and AP ($MEP_{AP}$) orientations at baseline. The same intensities were then reapplied 5 minutes and 30 minutes after the intervention, with 20 trials applied at each time point. While the I-wave specificity of these measures is generally suggested to rely on concurrent activation of the target muscle [27], post-intervention muscle activation is also known to strongly influence neuroplasticity induction [29–31]. As the current study was primarily concerned with plasticity induction, these measures were therefore applied in a resting muscle in order to minimize confounding effects of voluntary contraction. Given the likely independence of the intracortical circuits activated with different currents, these measures would still provide useful physiological insight. To assess activation selectivity in the resting muscle, we recorded the onset latencies from the MEP blocks of $MEP_{1mV}$, $MEP_{PA}$ and $MEP_{AP}$ in 17 subjects at baseline;

5 subjects were excluded as MEP onset in their EMG data was contaminated by stimulation artefacts.

**Cerebellar-brain inhibition (CBI).** The strength of CB's inhibitory influence on M1 was assessed using CBI, a stimulation protocol involving a conditioning stimulus applied to CB 5 ms prior to a test stimulus applied to M1 [5]. In accordance with previous literature, CB stimulation was applied using a double cone coil, with the centre of the coil located 3 cm lateral and 1 cm inferior to the inion, along the line joining the inion and the external auditory meatus of the right ear. The coil current was directed downward, resulting in an upward induced current. The intensity of CB stimulation was set at 70%MSO, but was reduced for the comfort of 11 participants (no lower than 60%MSO [32,33]), whereas M1 stimulation was set at $MEP_{1mV}$. The dual coil configuration of this measurement meant that each coil was directly connected to an individual Magstim $200^2$ stimulator. As removing the BiStim unit would result in an increase in stimulus strength, the $MEP_{1mV}$ intensity was checked prior to baseline CBI measures, and adjusted when required. Because antidromic activation of corticospinal neurons may confound measures of CBI [34], we ensured that the CB conditioning stimulus was at least 5% MSO below the active motor threshold for the corticospinal tract [35]. Only one participant demonstrated antidromic activation at 70%MSO. Measures of CBI were assessed at baseline, 5 minutes and 30 minutes post-intervention, with 15 trials recorded for each condition at each time point.

**I-wave periodicity repetitive TMS (iTMS).** In accordance with previous literature [19,36], iTMS consisted of 180 pairs of stimuli applied every 5 s, resulting in a total intervention time of 15 minutes. The same intensity was used for both stimuli, adjusted to produce a response of ~ 1mV when applied as a pair. These stimuli were applied using ISIs of 1.5 (I1; $iTMS_{1.5}$) and 4.5 ms (I3; $iTMS_{4.5}$) in separate sessions. These parameters produce robust potentiation of MEP amplitude [18,19,36,37]. A sham stimulation condition ($iTMS_{Sham}$) that was not expected to modulate corticospinal excitability was also applied in a third session. Within this condition, we stimulated intervals that corresponded to the transition between the peaks and troughs of facilitation that are observed within a complete SICF curve, as these were not expected to induce any changes in excitability. This included equal repetitions of 1.8, 2.3, 3.3, 3.8 and 4.7 ms ISI's, applied randomly and with an inter-trial jitter of 10%.

**Cerebellar Transcranial Direct Current Stimulation ($tDCS_{CB}$).** A Soterix Medical 1 x 1 DC stimulator (Soterix Medical, New York, NY) was used to apply tDCS to CB. Current was applied through saline-soaked sponge electrodes (EASYpads, 5 x 7 cm), with the cathode positioned over the same location used for CB TMS (i.e., 3 cm lateral and 1 cm inferior to inion, contralateral to M1 TMS) and anode positioned on the skin above the right Buccinator muscle [4,20,38]. Stimulation was applied at an intensity of 2 mA for 15 minutes [4,20,38], the same duration as the coincident application of iTMS to M1. Onset and offset of stimulation were ramped over a 30 s period prior to and following iTMS application.

## Data analysis

Analysis of EMG data was completed manually via visual inspection of offline recordings. For measures in resting muscle, any trials with EMG activity exceeding 25 μV in the 100 ms prior to stimulus application were excluded from analysis (across all participants, a total of 1.9% of trials were removed). All MEPs were measured peak-to-peak and expressed in mV. Onset latencies of the single-pulse MEP measures ($MEP_{1mV}$, $MEP_{PA}$ and $MEP_{AP}$) were assessed with a semi-automated process using a custom-written script within the Signal program (v 6.02, Cambridge Electronic Design) and expressed in ms. Onset of MEPs for each trial was defined as the point at which the rectified EMG signal following the stimulus artefact exceeded the

mean EMG amplitude plus 2 standard deviations within the 100 ms prior to the stimulus [39]. Measures of CBI were quantified by expressing the amplitude of individual trials produced by paired-pulse stimulation as a percentage of the mean response produced by single-pulse stimulation within the same block. For baseline measures of SICF, individual trials produced by paired-pulse stimulation were expressed as a percentage of the mean response produced by single-pulse stimulation within the same block. However, for post-intervention responses, previous work suggests that increased facilitation following iTMS correlates with the increased response to single pulse stimulation, and that this relationship cancels the effects of iTMS on SICF if the post-intervention single-pulse MEPs are used to normalise post-intervention SICF values [19]. As Spearman's rank correlation revealed a similar relationship within the data of the current study ($\rho = 0.7$, $P < 0.05$), individual post-intervention SICF trials were instead expressed relative to the mean pre-intervention single-pulse MEP [19]. For all TMS measures, effects of the intervention were quantified by expressing the post-intervention responses (normalised to the relevant single-pulse response for CBI and SICF) as a percentage of the pre-intervention responses.

## Statistical analysis

The distributions of the data residuals were visually inspected and assessed using Kolmogorov-Smirnov tests. These assessments indicated that the residuals were non-normal and positively-skewed. As the statistical analysis methods proposed within the protocol for this study [40] assume a normal distribution, we attempted to meet this assumption by applying log transformation. However, this failed to adequately adjust the data, and it was therefore necessary to identify an alternative test. Generalised linear mixed models (GLMM's), which are an extension of the linear mixed model analysis initially proposed, allow non-normal distributions to be accounted for [41]. We therefore elected to implement them within the current study. These were fitted with Gamma or Inverse Gaussian distributions [41], each model included single trial data with repeated measures, and all random subject effects were included (i.e., intercepts and slopes) [42]. Identity link functions were used for raw MEP responses and log link functions were used for MEPs expressed as a percentage (baseline SICF/CBI & baseline-normalised MEPs) [41,43]. In an attempt to optimise model fit, we tested different combinations of Gamma and Inverse Gaussian distributions with different covariance structures. The structure providing the best fit (assessed with the Schwartz Bayesian Criterion; BIC) within a model that was able to converge was used in the final model. To ensure measures were comparable between sessions, effects of iTMS session (iTMS$_{1.5}$, iTMS$_{4.5}$ & iTMS$_{Sham}$) on baseline measures of MEP$_{1mv}$, MEP$_{PA}$, MEP$_{AP}$ and CBI were investigated using one-factor GLMM, with each measurement investigated in a separate model. Furthermore, effects of iTMS session and ISI (1.5 & 4.5 ms) on baseline SICF were assessed using two-factor GLMM. Lastly, effects of iTMS session and TMS measure (MEP$_{1mV}$, MEP$_{PA}$ & MEP$_{AP}$) on baseline onset latencies were assessed using two-factor GLMM.

Changes in excitability during the intervention were assessed by comparing raw values taken from 18 blocks of 10 consecutive iTMS MEP trials between iTMS sessions, in addition to comparing normalised values taken from the first, middle and last 12 iTMS MEP trials between iTMS sessions. Changes in corticospinal excitability following the intervention were investigated by assessing the effects of iTMS session and time (5 minutes, 30 minutes) on baseline-normalised MEP$_{1mV}$ values. Changes in SICF measures of I-wave excitability following the intervention were investigated by assessing effects of iTMS session, time and ISI on baseline-normalised values. Changes in coil-orientation dependent measures of I-wave excitability following the intervention were investigated by assessing effects of iTMS session and time on

**Table 1. Baseline characteristics (mean ± STD) between iTMS sessions.**

| Characteristic | iTMS$_{1.5}$ | iTMS$_{4.5}$ | iTMS$_{Sham}$ |
|---|---|---|---|
| RMT (% MSO) | 44.1 ± 1.2 | 44.5 ± 1.5 | 44.5 ± 1.4 |
| MEP$_{1mV}$ (% MSO) | 55.4 ± 2.0 | 54.0 ± 1.9 | 56.0 ± 2.5 |
| MEP$_{PA}$ (% MSO) | 49.6 ± 1.6 | 49.0 ± 1.5 | 50.0 ± 1.9 |
| MEP$_{AP}$ (% MSO) | 65.5 ± 1.6 | 65.4 ± 2.0 | 66.8 ± 2.2 |
| iTMS (% MSO) | 48.7 ± 2.0 | 49.2 ± 1.7 | 49.3 ± 2.0 |

baseline-normalised values separately for MEP$_{PA}$ and MEP$_{AP}$. Changes in CBI following the intervention were investigated by assessing the effects of iTMS session and time on baseline-normalised CBI values. For all models, investigation of main effects and interactions were performed using custom contrasts with Bonferroni correction, and significance was set at $P < 0.05$. Data for all models are presented as estimated marginal means, whereas pairwise comparisons are presented as the estimated mean difference (EMD) and 95% confidence interval (95%CI) for the estimate, providing a non-standardised measure of effect size.

Spearman's rank correlation was used to investigate interactions between variables. Specifically, changes in CBI due to the intervention were correlated with changes in measures of corticospinal and intracortical function in order to assess if alterations within the CB-M1 pathway contributed to plasticity effects. In addition, changes in intracortical function due to the intervention were correlated with changes in corticospinal function in an attempt to identify if generalised changes in excitability were driven by alterations within specific circuits. Multiple comparisons within these analyses were corrected using a Bonferroni adjustment (significance set at $P < 0.0019$).

## Results

All participants completed the experiment in full and without adverse reactions. At baseline, there was no difference between sessions for RMT or the stimulus intensities required to produce MEP$_{1mV}$, MEP$_{PA}$ or MEP$_{AP}$ (Table 1). Baseline measures of corticospinal, intracortical, and cerebellar excitability are shown in Table 2. There was no significant difference between sessions for MEP$_{1mV}$ ($F_{2,1300} = 2.6$, $P = 0.08$), MEP$_{PA}$ ($F_{2,1310} = 2.0$, $P = 0.1$) and MEP$_{AP}$ ($F_{2,1290} = 0.0$, $P = 1$). SICF varied between ISIs ($F_{1,1560} = 35.5$, $P < 0.05$), with *post-hoc* comparisons showing reduced facilitation for SICF$_{4.5}$ relative to SICF$_{1.5}$ (EMD = 43.9 [28.6, 59.1], $P < 0.05$). However, there was no difference between sessions ($F_{2,1560} = 0.1$, $P = 0.9$) or interaction between factors ($F_{2,1560} = 0.0$, $P = 1$). There was no significant difference between

**Table 2. Baseline responses of corticospinal, intracortical and cerebellar excitability between iTMS sessions.**

| TMS protocol | | iTMS$_{1.5}$ | iTMS$_{4.5}$ | iTMS$_{Sham}$ |
|---|---|---|---|---|
| MEP$_{1mV}$ (mV) | | 0.90 [0.84, 0.96] | 1.00 [0.93, 1.07] | 0.92 [0.86, 0.99] |
| MEP$_{PA}$ (mV) | | 0.54 [0.50, 0.58] | 0.48 [0.44, 0.52] | 0.50 [0.46, 0.53] |
| MEP$_{AP}$ (mV) | | 0.50 [0.46, 0.54] | 0.50 [0.46, 0.53] | 0.49 [0.45, 0.53] |
| SICF (% test) | 1.5ms | 162.2 [137.4, 191.4] | 156.3 [132.4, 184.5] | 153.2 [129.9, 180.9] |
| | 4.5ms | 114.7 [97.2, 135.4][a] | 114.1 [96.7, 134.7][a] | 111.1 [94.1, 131.2][a] |
| CBI (% test) | | 74.6 [59.7, 93.1] | 66.4 [53.2, 82.8] | 70.1 [56.8, 88.6] |
| iTMS first epoch | | 1.06 [0.86, 1.27] | 0.82 [0.63, 1.01] | 0.78 [0.59, 0.96] |

Data show mean [95%CI; lower, upper].

[a]$P < 0.05$ compared to SICF1.5 within the same condition.

**Table 3. Baseline onset latencies of corticospinal (MEP$_{1mV}$) and intracortical (MEP$_{PA}$ and MEP$_{AP}$) measures between iTMS sessions.**

| Characteristic | iTMS$_{1.5}$ | iTMS$_{4.5}$ | iTMS$_{Sham}$ |
|---|---|---|---|
| MEP$_{1mV}$ (ms) | 21.6 [20.7, 22.5][a] | 21.6 [20.7, 22.5][a] | 21.6 [20.7, 22.5][a] |
| MEP$_{PA}$ (ms) | 21.8 [20.9, 22.7][a] | 21.7 [20.7, 22.6][a] | 22.0 [21.0, 22.9][a] |
| MEP$_{AP}$ (ms) | 22.9 [22.0, 23.9] | 22.8 [21.9, 23.8] | 22.8 [21.9, 23.8] |

Data show mean [95%CI; lower, upper].

[a]$P < 0.05$ compared to MEP$_{AP}$.

sessions for CBI ($F_{2,950} = 0.7$, $P = 0.5$). In contrast, responses recorded during the first epoch of iTMS varied between conditions ($F_{2,657} = 3.0$, $P < 0.05$), but no differences were found with *post-hoc* testing (all $P > 0.05$). Onset latencies for the single-pulse MEP measures (MEP$_{1mV}$, MEP$_{PA}$ & MEP$_{AP}$) are shown in Table 3. These varied between TMS measures ($F_{2,2950} = 50.6$, $P < 0.05$), with MEP$_{AP}$ showing longer latencies than both MEP$_{PA}$ (EMD = 1.0 [0.8, 1.3], $P < 0.05$) and MEP$_{1mV}$ (EMD = 1.3 [0.9, 1.6], $P < 0.05$). There was no significant difference between sessions ($F_{2,2950} = 0.3$, $P = 0.7$) or interaction between factors ($F_{4,2950} = 1.2$, $P = 0.3$).

## Corticospinal excitability during the intervention

Fig 2A shows changes in MEP amplitude during iTMS, presented as 18 epochs of 10 trials. These values varied between sessions ($F_{2,11600} = 13.7$, $P < 0.05$), with *post-hoc* comparisons showing increased MEP amplitude for iTMS$_{1.5}$ compared to iTMS$_{4.5}$ (EMD = 0.2 [0.1, 0.3]; $P < 0.05$) and iTMS$_{Sham}$ (EMD = 0.3 [0.2, 0.4]; $P = 0.003$), and increased MEP amplitude for iTMS$_{4.5}$ relative to iTMS$_{Sham}$ (EMD = 0.1 [0.0, 0.2]; $P = 0.04$). There was also a difference between iTMS blocks ($F_{17,11600} = 1.7$, $P = 0.03$), but no differences were found with *post-hoc* testing (all $P > 0.05$). There was no interaction between factors ($F_{34,11600} = 1.1$, $P = 0.4$). As responses during the first block of iTMS varied between sessions (despite no significant results), values taken from the middle and last 12 consecutive MEP trials were normalised to the average amplitude of the first 12 trials (Fig 2B). These values did not differ between sessions ($F_{2,1580} = 0.3$, $P = 0.7$) or time points ($F_{1,1580} = 0.1$, $P = 0.8$) and there was no interaction between factors ($F_{2,1580} = 2.4$, $P = 0.09$; Fig 2B).

## Post-intervention changes in corticospinal excitability, intracortical excitability and CBI

**Corticospinal excitability.** Changes in MEP$_{1mV}$ following the intervention are presented in Fig 3. There was no difference between sessions ($F_{2,2580} = 2.2$, $P = 0.1$) or time ($F_{1,2580} = 0.6$, $P = 0.5$), and no interaction between factors ($F_{2,2580} = 0.0$, $P = 1$).

**I-wave excitability.** Fig 4 shows changes in SICF$_{1.5}$ (Fig 4A) and SICF$_{4.5}$ (Fig 4B) following the intervention. There was no difference between sessions ($F_{2,3110} = 1.9$, $P = 0.1$), time ($F_{1,3110} = 0.8$, $P = 0.4$), or ISIs ($F_{1,3110} = 1.5$, $P = 0.2$), and no interaction between factors (all $P > 0.05$).

Fig 5 shows changes in MEP$_{PA}$ (Fig 5A) and MEP$_{AP}$ (Fig 5B) following iTMS. For MEP$_{PA}$, data varied between sessions ($F_{2,2580} = 4.0$, $P = 0.02$; Fig 5A), with *post-hoc* comparisons showing increased responses following both iTMS$_{1.5}$ (EMD = 35.2 [0.1, 70.4]; $P < 0.05$) and iTMS$_{4.5}$ (EMD = 34.1 [0.1, 68.2]; $P < 0.05$) relative to iTMS$_{Sham}$. However, there was no difference between time points ($F_{1,2580} = 0.4$, $P = 0.5$) and no interaction between factors ($F_{2,2580} = 1.5$, $P = 0.2$). For MEP$_{AP}$, there was no difference between sessions $F_{2,2590} = 0.3$, $P = 0.7$; Fig 5B) or time ($F_{1,2590} = 0.2$, $P = 0.6$), and no interaction between factors ($F_{2,2590} = 0.7$, $P = 0.5$).

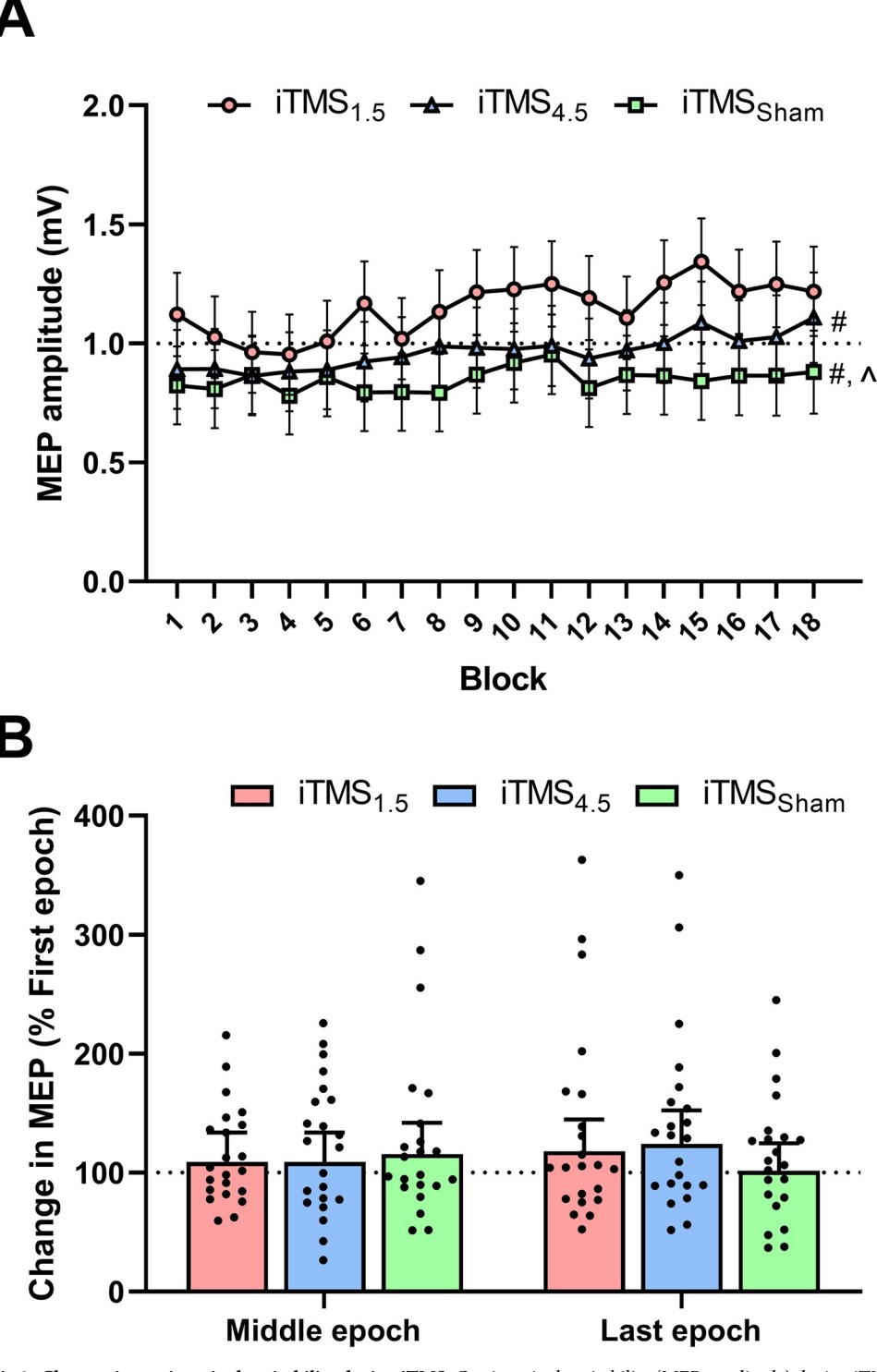

**Fig 2. Changes in corticospinal excitability during iTMS.** Corticospinal excitability (MEP amplitude) during $iTMS_{1.5}$ (red circles/bars), $iTMS_{4.5}$ (blue triangles/bars), and $iTMS_{Sham}$ (green squares/bars) are presented as raw values averaged over 10 consecutive MEP trials, resulting in 18 blocks (A), or as the estimated normalised values of the middle and last blocks of 12 consecutive MEP trials, expressed relative to the average response of the first 12 trials. (B). #$P < 0.05$ session comparison to $iTMS_{1.5}$. $\wedge P < 0.05$ session comparison to $iTMS_{4.5}$.

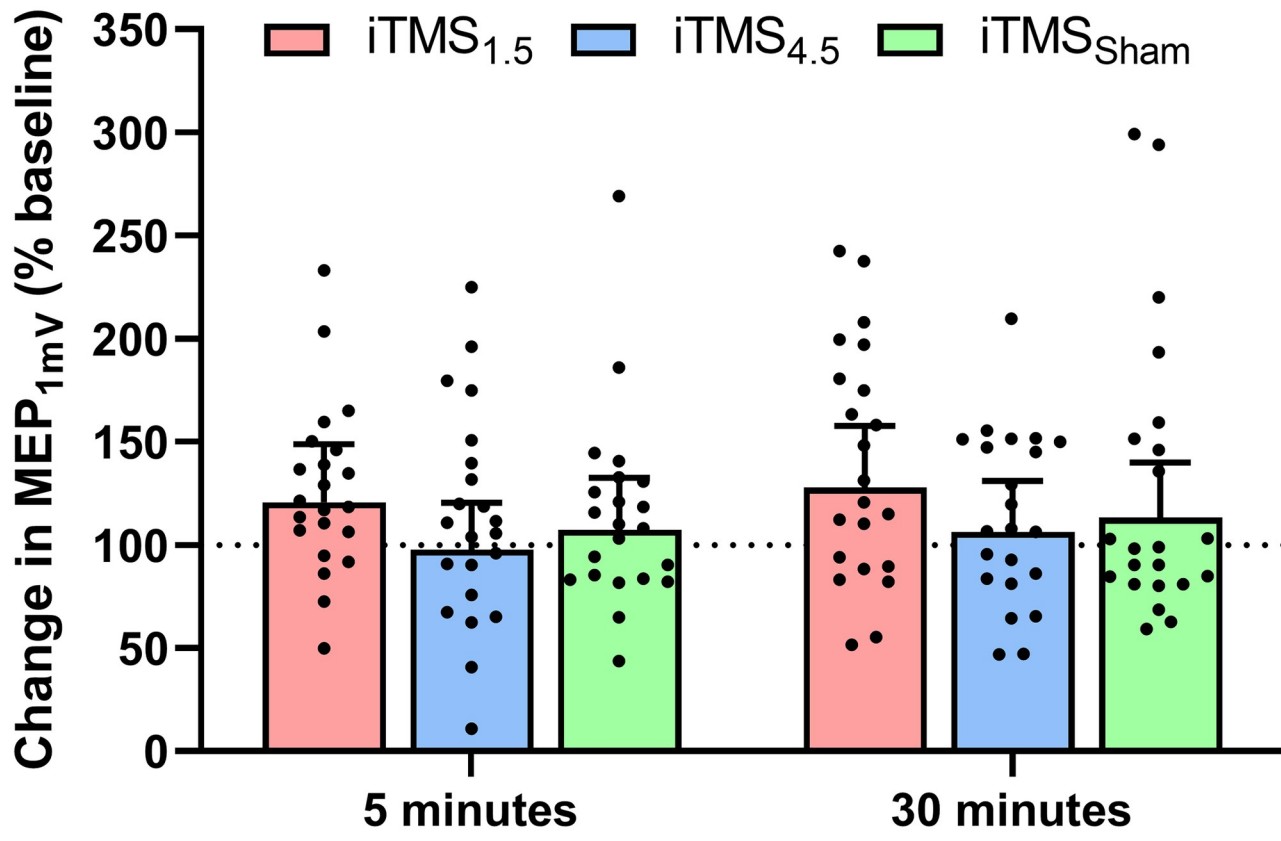

**Fig 3. Changes in corticospinal excitability following iTMS$_{1.5}$ (red), iTMS$_{4.5}$ (blue), and iTMS$_{Sham}$ (green) at 5 and 30 minutes.**

**Cerebellar excitability.** Changes in CBI following the intervention are presented in Fig 6. CBI did not differ between sessions ($F_{2,1920} = 0.4$, $P = 0.7$) or time points ($F_{1,1920} = 0.1$, $P = 0.8$), and there was no interaction between factors ($F_{2,1920} = 0.7$, $P = 0.5$).

## Correlation analyses

Changes in CBI were not related to changes in corticospinal (MEP$_{1mV}$) or intracortical (SICF$_{1.5}$, SICF$_{4.5}$, MEP$_{PA}$ & MEP$_{AP}$) function following the intervention (all $P > 0.019$). In contrast, changes in MEP$_{PA}$ predicted changes in MEP$_{1mV}$ following and iTMS$_{Sham}$ ($\rho = 0.7$, $P < 0.019$), but not iTMS$_{1.5}$ ($\rho = 0.4$, $P = 0.06$) or iTMS$_{4.5}$ ($\rho = 0.4$, $P = 0.04$). Changes in MEP$_{AP}$ predicted changes in MEP$_{1mV}$ following iTMS$_{Sham}$ ($\rho = 0.7$, $P < 0.019$), but not iTMS$_{1.5}$ ($\rho = 0.4$, $P = 0.04$) or iTMS$_{4.5}$ ($\rho = 0.5$, $P = 0.01$). Changes in SICF$_{1.5}$ predicted changes in MEP$_{1mV}$ following and iTMS$_{Sham}$ ($\rho = 0.7$, $P = 0.001$), but not iTMS$_{1.5}$ ($\rho = 0.6$, $P = 0.006$) or iTMS$_{4.5}$ ($\rho = 0.4$, $P = 0.09$). Changes in SICF$_{4.5}$ predicted changes in MEP$_{1mV}$ following iTMS$_{1.5}$ ($\rho = 0.7$, $P = 0.001$), but not iTMS$_{4.5}$ ($\rho = 0.2$, $P = 0.4$) or iTMS$_{Sham}$ ($\rho = 0.6$, $P = 0.002$).

## Discussion

The present study assessed how changes in CB activity influence the excitability and neuro-plastic response of I-wave-producing circuits in M1. This was achieved by modulating CB excitability with cathodal tDCS$_{CB}$ while concurrently inducing plastic changes in M1 with iTMS. Corticospinal excitability was assessed during the intervention and changes in

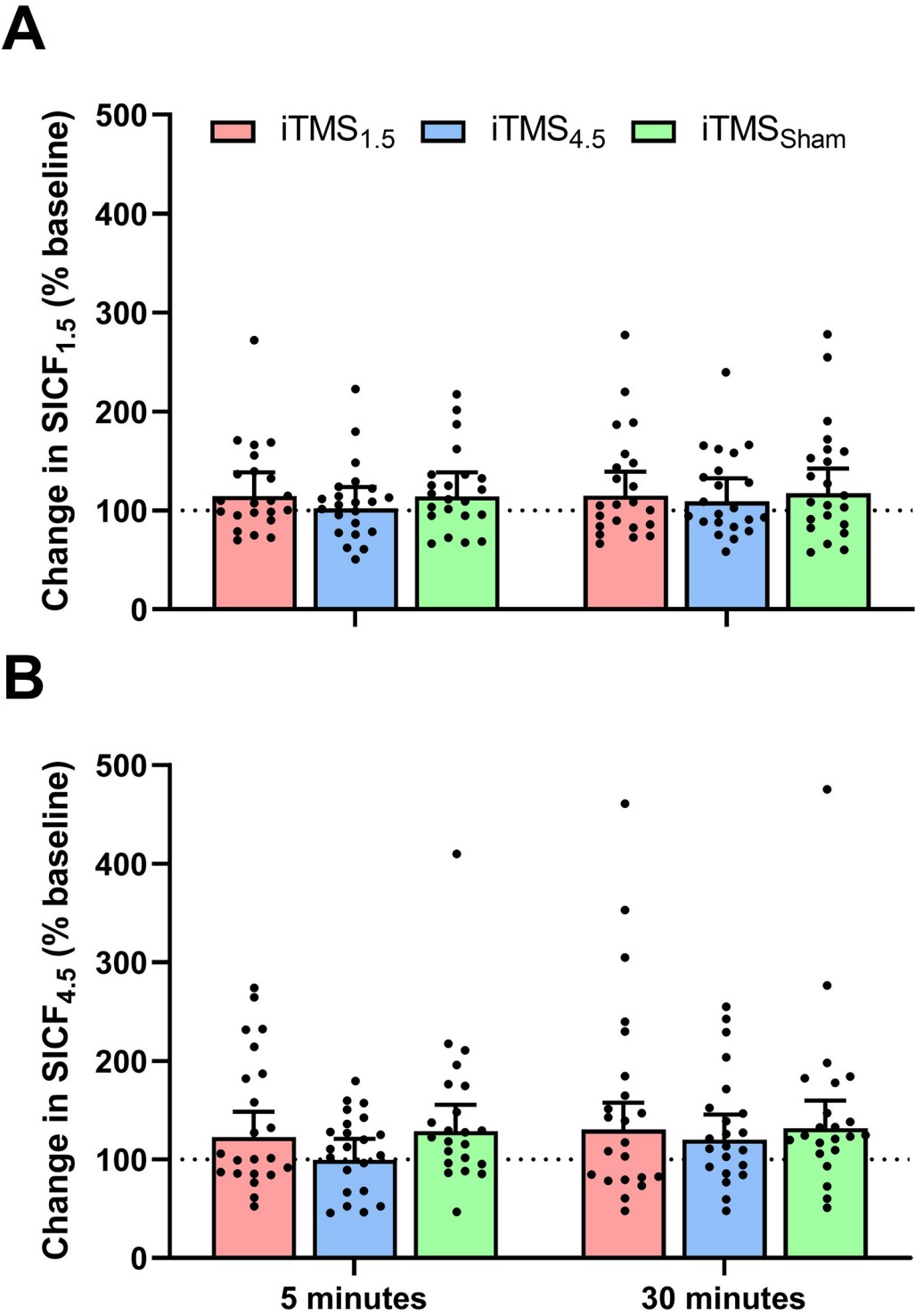

**Fig 4. Changes in SICF$_{1.5}$ (A) and SICF$_{4.5}$ (B) following iTMS$_{1.5}$ (red), iTMS$_{4.5}$ (blue), and iTMS$_{Sham}$ (green) at 5 and 30 minutes.**

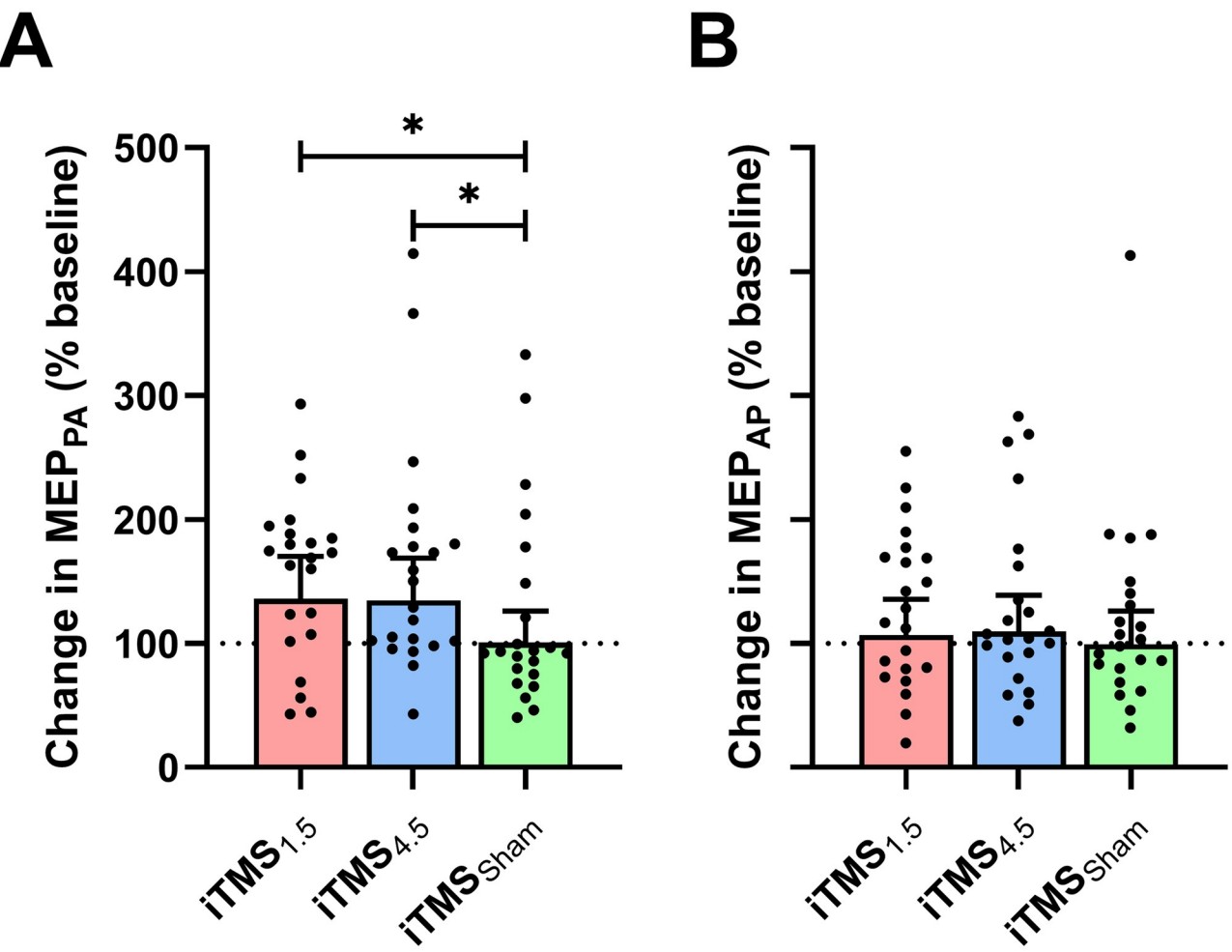

**Fig 5. Changes in MEP$_{PA}$ (A) and MEP$_{AP}$ (B) following iTMS$_{1.5}$ (red), iTMS$_{4.5}$ (blue), and iTMS$_{Sham}$ (green).** *$P < 0.05$.

corticospinal excitability (MEP$_{1mV}$), intracortical excitability (SICF, MEP$_{PA}$ & MEP$_{AP}$) and CBI were assessed post-intervention. During the intervention, facilitation expected from iTMS (when applied in isolation) was reduced (iTMS$_{1.5}$) or removed (iTMS$_{4.5}$). Furthermore, although MEP$_{PA}$ was potentiated following both iTMS$_{1.5}$ and iTMS$_{4.5}$, no other measure of corticospinal, intracortical or CB-M1 excitability was altered after the intervention.

## CB influence on corticospinal excitability

Previous work has shown that iTMS applied in isolation results in a 150–500% increase in MEP amplitude during the intervention, with potentiation of ~ 150–400% persisting for up to 15 minutes post-intervention [18,19,44]. This facilitatory response is thought to be mediated by the neuroplastic reinforcement of trans-synaptic events involving early (I1; iTMS$_{1.5}$) and late (I3; iTMS$_{4.5}$) I-wave circuits [18]. In contrast, MEP$_{1mV}$ both during and after the intervention of the current study did not vary over time, indicating that iTMS was unable to modify corticospinal excitability. Consequently, our findings suggest that the coincident application of cathodal tDCS$_{CB}$ appears to disrupt neuroplastic changes in corticospinal excitability following iTMS. This outcome is consistent with the findings of a previous study, which reported that

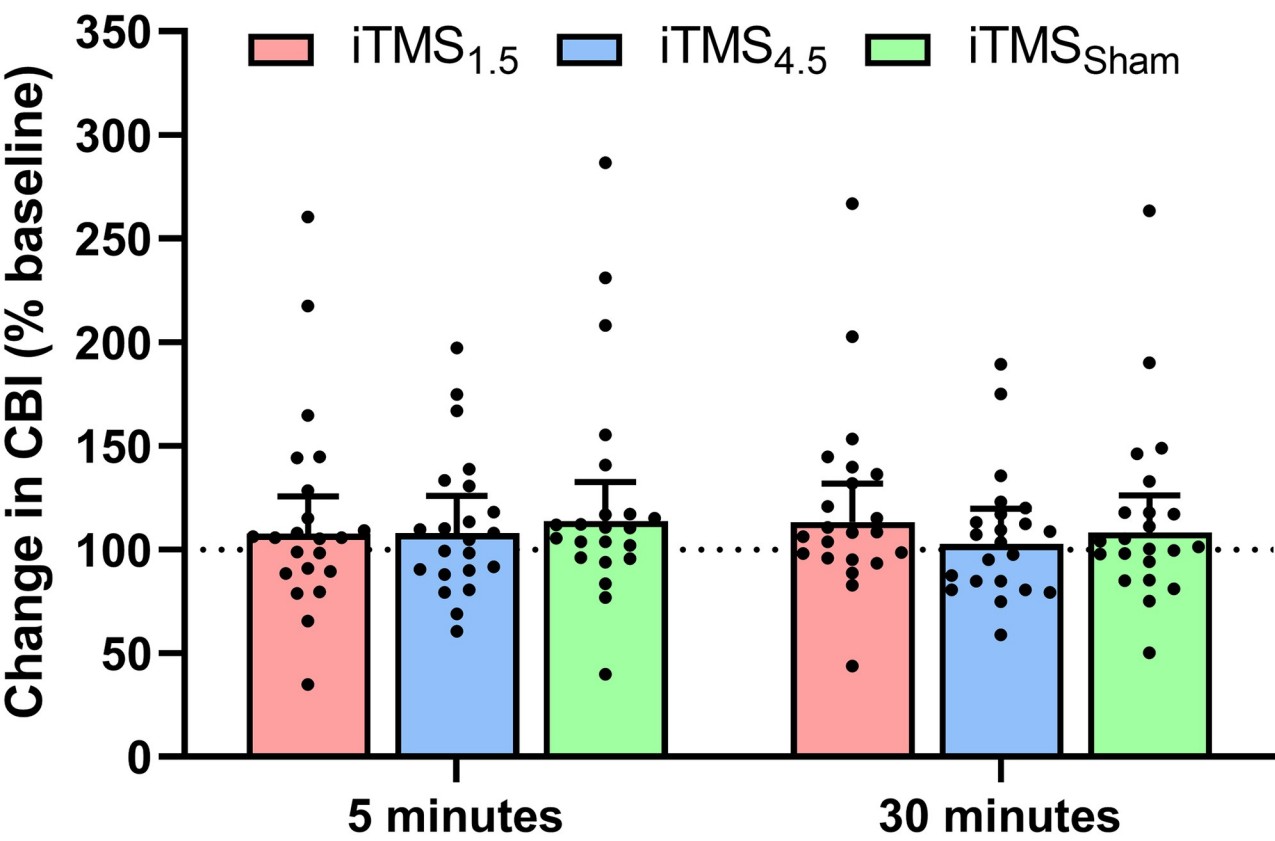

**Fig 6. Changes in CBI following iTMS$_{1.5}$ (red), iTMS$_{4.5}$ (blue), and iTMS$_{Sham}$ (green) at 5 and 30 minutes.**

the response to paired-associative stimulation (PAS, an alternative plasticity paradigm) is abolished by coincident cathodal tDCS$_{CB}$ [38]. However, the effects in that study were suggested to stem from tDCS$_{CB}$ interfering with transcerebellar sensory inputs to M1. Given that iTMS-induced plasticity does not rely on sensory inputs that are critical for PAS, the disruption observed in the current study is likely mediated by a different mechanism. One possibility is that cathodal tDCS$_{CB}$ reduced Purkinje cell excitability [20,45], resulting in disinhibition of dento-thalamo-cortical projections to M1 and a subsequent shift in local excitability that influenced the response to iTMS. In particular, metaplasticity mechanisms that can remove or reverse the response to a given plasticity intervention (based on recent synaptic activity) are well-documented within M1 [46–48]. It is therefore plausible that disinhibition of the dento-thalamo-cortical pathway may have resulted in a metaplastic response to iTMS that removed the expected facilitation of MEP$_{1mV}$.

While cathodal tDCS$_{CB}$ appeared to reduce the response to both iTMS$_{1.5}$ and iTMS$_{4.5}$, the magnitude of this effect varied between conditions. In particular, responses during the iTMS$_{4.5}$ intervention were significantly reduced relative to iTMS$_{1.5}$. Although this could be suggested to reflect differential effects of tDCS$_{CB}$ on early and late I-wave circuits, interpretation of this data is complicated by the numerical differences in MEP amplitude at the start of iTMS (see Fig 2A and Table 2). While these were not significant, their nature mirrored differences between conditions during the intervention, and may therefore have confounded the response to iTMS. Indeed, when the analysis was repeated using data that were normalised to the first iTMS block, differences between conditions were removed (Fig 2B). Consequently, the effects

of tDCS$_{CB}$ on the corticospinal response to iTMS do not appear to vary between interventions targeting different I-wave circuits.

## CB influence on intracortical excitability

Previous work applying iTMS in isolation reported increased SICF post-intervention [19,44]. Specifically, iTMS$_{1.5}$ has been demonstrated to potentiate SICF at ISIs of 1.5 ms and 4.5 ms [19], whereas iTMS$_{4.5}$ has been shown to potentiate SICF at ISIs of 4–5 ms [44]. Given that SICF is thought to index excitability within the intracortical circuits responsible for I-wave generation [21], these findings have been suggested to reflect neuroplastic effects of iTMS within these circuits. In contrast, our results failed to demonstrate this expected increase following iTMS. One interpretation of this outcome is that cathodal tDCS$_{CB}$ broadly disrupted the neuroplastic response of intracortical networks to iTMS, similar to our findings for the measures of corticospinal excitability (see above). However, a limitation of this interpretation is that SICF was also unchanged within the iTMS$_{Sham}$ condition (which still included real tDCS$_{CB}$), despite previous work showing that SICF is modified by tDCS$_{CB}$ applied in isolation [15]. Consequently, it is currently unclear if the negative response of SICF to our intervention reflects an interaction between cortical and CB stimulation (e.g., disfacilitation of intracortical circuits), or indicates reduced sensitivity of SICF measures within the current study. Despite this, it is important to note that changes in SICF that were previously reported following tDCS$_{CB}$ were identified using threshold tracking paired-pulse TMS [15]; given that this technique may have greater sensitivity for identifying changes in intracortical excitability following an intervention [49,50], reduced sensitivity appears the more likely explanation. It will be important to clarify this limitation in future work using alternative measures of intracortical networks that are more sensitive to changes in CB activity.

Measures of MEP$_{PA}$ and MEP$_{AP}$ were included as alternative indices of intracortical excitability. When applied at low intensities, the conventional interpretation of these measures has been that they reflect activity in early (I1) and late (I3) I-wave circuits, respectively [13]. However, while it is clear that this methodological approach can selectively recruit different I-wave volleys, growing evidence suggests that these likely originate from different intracortical populations (e.g., PA and AP late I-waves are generated by non-identical intracortical circuits; [13,16,51,52]). Within the current study, while MEP$_{PA}$ was potentiated by both iTMS interventions, MEP$_{AP}$ was unaffected by either. Two conclusions can be drawn from this outcome. First, it is unlikely that our measures of MEP$_{PA}$ were able to isolate early I-wave activity: if responses were selective to the early I-waves, a facilitatory response to iTMS$_{4.5}$ would not be expected. This limitation likely stemmed from the increased stimulation intensity required to record these measures in a resting muscle, whereas I-wave selectivity is generally increased by applying stimulation during muscle activation [27]. It also likely explains why MEP$_{PA}$ and MEP$_{1mV}$ had similar onset latency values (Table 3). However, our use of a resting state was a deliberate decision intended to avoid the confounding influence that muscle activation can have on neuroplasticity induction [53].

Second, this pattern of response—simultaneously suggesting that late I-waves were (MEP$_{PA}$) and were not (MEP$_{AP}$) responsive to iTMS—appears to be consistent with the more contemporary interpretation that different current directions recruit from different intracortical populations (for review, see; [54]). As previous work has shown that AP MEPs are potentiated by iTMS in isolation [44], our results could therefore suggest that tDCS$_{CB}$ influenced (reduced) the neuroplastic response of intracortical circuits activated by AP stimulation. However, an important limitation to this explanation is that the reliability of AP-responses to iTMS has not been well-established. In particular, previous effects of iTMS$_{4.5}$ on MEP$_{AP}$ have been

relatively weak [44], while potentiation of AP responses following iTMS$_{1.5}$ has not been previously demonstrated. Consequently, we cannot exclude the possibility that reduced efficacy of iTMS within AP circuits contributed to the lack of MEP$_{AP}$ facilitation. One factor that may have influenced this outcome was our application of iTMS with a PA current. It will therefore be interesting for future work to assess how the response to AP iTMS interacts with the cerebellum.

As neuroplastic changes in MEP$_{1mV}$ and SICF (i.e., measures recorded with PA stimulation) were also absent, it may appear counterintuitive to suggest that tDCS$_{CB}$ targets AP circuits. However, although different current directions can be expected to selectively target different interneuronal networks at sufficiently low stimulus intensities, there will be greater overlap between recruited populations as stimulus intensity increases [44]. Given that MEP$_{1mV}$ and SICF were recorded with higher stimulus intensities than were used for MEP$_{PA}$, we could therefore speculate that the differential response between these measures was driven by MEP$_{1mV}$ and SICF having a greater relative contribution from intracortical circuits that are AP-sensitive at low intensity, but able to be recruited by PA currents as intensity increases. Furthermore, the kaleidoscope of significant correlations we observed between variables may be also partially explained by the mixed recruitment of different intracortical populations. While yet to be verified in future work, this outcome nonetheless illustrates the importance of stimulus intensity for study design and interpretation, particularly with respect to I-wave circuits.

## Changes to CBI

Previous work applying cathodal tDCS$_{CB}$ in isolation reported reduced CBI following the intervention [20,45]. This reduction is thought to be mediated by the downregulation of Purkinje cell excitability, resulting in disinhibition of the dento-thalamo-cortical pathway [20,45]. While characteristics of baseline CBI within the current study were comparable to a previous study that reported this reduction [45], we were still unable to demonstrate any changes in CBI following the intervention; an outcome that is particularly surprising for iTMS$_{Sham}$, which still involved real tDCS$_{CB}$. While the current study is unable to provide any experimental data to clarify this lack of modulation, our coincident application of stimuli over both M1 and CB may offer some explanation. Specifically, CB-M1 connectivity is bidirectional, with projections from M1 to CB mediated by the cortico-ponto-cerebellar pathway [55,56]. Although functional investigation of this pathway has been limited to stroke patients, these connections seem to be related to performance of fine motor skills [57]. Furthermore, animal studies have demonstrated that motor and somatosensory activity is closely related to activation of mossy fibres [58,59], which influence CB processing [55]. It may therefore be possible that stimulation of M1 (even during iTMS$_{Sham}$) resulted in a reciprocal disruption of CB neuroplastic response to cathodal tDCS$_{CB}$. However, further work is required to characterise the physiology of these connections.

The absence of additional control conditions in which real iTMS was applied over M1 in conjunction with sham tDCS$_{CB}$ is a limitation of the current study. Given that the isolated response of M1 to iTMS targeting early (I1) and late (I3) I-waves has been established previously, we decided to omit this condition in order to minimise the number of experimental sessions for each participant. However, our inability to replicate these responses within the current cohort limits the conclusions we are able to draw. It will therefore be important for future replication of these results to also include sessions that apply real iTMS in conjunction with sham tDCS$_{CB}$. Another limitation is that we specifically targeted the I3-wave, despite both I2- and I3-waves being considered late I-waves. This design was based on the large body of

existing evidence demonstrating the physiological and functional importance of I3-waves. However, some evidence suggests that CB may also interact with the I2-wave [15], and this possibility should therefore be assessed in future work.

In conclusion, the application of cathodal tDCS$_{CB}$ disrupted the neuroplastic effects of iTMS on corticospinal and intracortical excitability. Importantly, our results provide preliminary evidence that this effect may be selectively mediated by AP-sensitive circuits. However, further work involving additional sham stimulation conditions, as well as measures more sensitive to the specific circuits targeted by CB, will be required to confirm this mechanism.

## Author Contributions

**Conceptualization:** George M. Opie.

**Data curation:** Wei-Yeh Liao.

**Formal analysis:** Wei-Yeh Liao.

**Funding acquisition:** George M. Opie.

**Investigation:** Wei-Yeh Liao, Ryoki Sasaki.

**Methodology:** Ryoki Sasaki, John G. Semmler, George M. Opie.

**Project administration:** George M. Opie.

**Resources:** George M. Opie.

**Supervision:** John G. Semmler, George M. Opie.

**Visualization:** Wei-Yeh Liao.

**Writing – original draft:** Wei-Yeh Liao.

**Writing – review & editing:** Wei-Yeh Liao, Ryoki Sasaki, John G. Semmler, George M. Opie.

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
