## [Decision Letter · Decision Letter 0]

9 Mar 2022

PONE-D-22-03125Cerebellar transcranial direct current stimulation disrupts neuroplasticity of intracortical motor circuits.PLOS ONE

Dear Dr. Opie,

Thank you for submitting your manuscript to PLOS ONE. After careful consideration, we feel that it has merit but does not fully meet PLOS ONE’s publication criteria as it currently stands. Therefore, we invite you to submit a revised version of the manuscript that addresses the points raised during the review process.

Two experts reviewed your manuscript. As you will see below, both reviewers have questions regarding the methodological approach. However, since your paper was based on a previously registered protocol in PLoS one,  I suggest you examine these questions to see if you can provide some justification for the reviewers. Otherwise, you can focus your attention on the reviewers' suggestions about the interpretation of the results and their significance for the field.

We look forward to receiving your revised manuscript.

Kind regards,

François Tremblay, PhD

Academic Editor

PLOS ONE

Journal Requirements:

3. Thank you for submitting the above manuscript to PLOS ONE. During our internal evaluation of the manuscript, we found significant text overlap between your submission and the following previously published work:

- https://journals.plos.org/plosone/article?id=10.1371%2Fjournal.pone.0236005

Please revise the manuscript to rephrase the duplicated text, cite your sources, and provide details as to how the current manuscript advances on previous work. Please note that further consideration is dependent on the submission of a manuscript that addresses these concerns about the overlap in text with published work.

Reviewers' comments:

Reviewer's Responses to Questions

**Comments to the Author**

1. Does the manuscript adhere to the experimental procedures and analyses described in the Registered Report Protocol?

If the manuscript reports any deviations from the planned experimental procedures and analyses, those must be reasonable and adequately justified.

Reviewer #1: Yes

Reviewer #2: No

2. If the manuscript reports exploratory analyses or experimental procedures not outlined in the original Registered Report Protocol, are these reasonable, justified and methodologically sound?

A Registered Report may include valid exploratory analyses not previously outlined in the Registered Report Protocol, as long as they are described as such.

Reviewer #1: Yes

Reviewer #2: Partly

3. Are the conclusions supported by the data and do they address the research question presented in the Registered Report Protocol?

The manuscript must describe a technically sound piece of scientific research with data that supports the conclusions. The conclusions must be drawn appropriately based on the research question(s) outlined in the Registered Report Protocol and on the data presented.

Reviewer #1: Yes

Reviewer #2: Partly

4. Have the authors made all data underlying the findings in their manuscript fully available?

Reviewer #1: Yes

Reviewer #2: Yes

5. Is the manuscript presented in an intelligible fashion and written in standard English?

Reviewer #1: Yes

Reviewer #2: Yes

6. Review Comments to the Author

Please use the space provided to explain your answers to the questions above. (Please upload your review as an attachment if it exceeds 20,000 characters)

Reviewer #1: In this article written by Liao et al., the authors present the results of registered result that investigated whether applying cathodal tDCS over the cerebellum influences the activity of intracortical neurons within the primary motor cortex. To do this, the authors made use of a technique, termed iTMS, that produces neuromodulatory affects over M1 when applied at specific intervals that represent early and late indirect inputs on to corticospinal neurons, and assessed whether the changes in M1 excitability following iTMS is disrupted with cathodal tDCS. The article is easy to follow as it is very well-written, and from a technical stand-point, it appears the authors did a good job of reliably conducting some quite complex physiological measures. I have some issues with some of the selected produces conducted in their experiments that require justification and with some of the result interpretations that could be revised, but the overall quality of this manuscript is quite high. Considering this is a registered study, I think the results carry a lot of weight, especially when considering that these types of reports are not easy to find in the literature of brain stimulation. Below are my concerns:

For the iTMS protocol, can the authors justify why they selected to target the I3 wave (at 4.5 ms) as opposed to (or in addition to) selecting an interval that would predominately target the I2 wave? Since both the I2 and I3 wave get lumped together as “late I-waves”, should we expect a similar finding if one were to target the I2 wave?

“The absence of additional control conditions in which real iTMS was applied over M1 in conjunction with sham tDCSCB is a limitation of the current study.” The problem with not conducting this additional control condition is that we don’t really know the reliability and reproducibility of iTMS. For instance, what do the authors believe is the error of margin of using the ISI of 4.5 ms for targeting the I3 wave? For instance, Long et al., 2017 ( 10.1093/brain/awx102) used an ISI of 4.3 ms for their late I-wave protocol, and the authors of this manuscript previously show that an ISI of 4 ms and 5ms yielded greater facilitation on M1 excitability when compared to 4.5ms. Thus, it seems that another potential interpretation of the results could be that the lack of changes on intracortical excitability and corticospinal excitability might be due to the variability of the “late” iTMS protocol and not influences from the cerebellum.

Why did the authors select cathodal cerebellar Tdcs as their cerebellar neuromodulatory technique as opposed to cerebellar theta burst protocol? Cerebellar tDCS does appear to exclusively produces changes in cerebellar excitability and not corticospinal activity (Galea et al., 2009; 10.1523/JNEUROSCI.2184-09.2009), which has been suggested to produce tonic changes at the Purkinje cell level without activating cerebellar-thal-M1 pathways. However, cerebellar theta burst does yield changes in M1 excitability (10.1016/j.clinph.2008.08.008 ; 10.1038/srep36191), which might reflect this protocol depolarizing cerebellar Purkinje cells and engaging cerebellar-thal-M1 pathways (that produce changes in M1).Thus, one could presume that cerebellar theta burst would have yielded a more disruptive affect on the iTMS protocol.

“Second, this pattern of response – simultaneously suggesting that late I-waves were (MEPPA) and were not (MEPAP) responsive to iTMS – can only be consistent with the more contemporary interpretation that different current directions recruit from different intracortical populations (for review, see; [54]). As previous work has shown that AP MEPs are potentiated by iTMS in isolation [44], our results therefore suggest that tDCSCB influenced (reduced) the neuroplastic response of intracortical circuits activated by AP stimulation.”

While I agree with that different currents are likely recruiting different population of neurons, the previous results by the authors demonstrating changes in AP MEPs following iTMS performed with PA currents (ref 44) does not seem to exactly fit with this notion. One would rather expect that iTMS with PA currents produces greater facilitation of PA MEPs than AP MEPs if distinct populations are being targeted with directional TMS. Thus, if the authors are suggesting that cerebellar tDCS targets AP circuits, would it not be more reasonable to perform iTMS with AP currents?

“SICF utilised a conditioning stimulus set at 90% RMT, a test stimulus set at MEP1mV and two ISIs of 1.5 (SICF1.5) and 4.5 (SICF4.5) ms, which correspond to the early and late MEP peaks apparent in a complete SICF curve [22, 26, 27]. Measurements of SICF included 12 trials for each condition, at each time point”

Why did the authors select only 12 trials for each SICF measure, as opposed to 15 trials for each CBI measure and 20 trials for 1 mV assessment? Is SICF considered more reliable than these other measures to justify the much smaller sample?

“The intensity of CB stimulation was set at 70%MSO, but was at times reduced for participant comfort (no lower than 60%MSO [33]), whereas M1 stimulation was set at MEP1mV” ; “Because antidromic activation of corticospinal neurons may confound measures of CBI [34], we ensured that the CB conditioning stimulus was at least 5% MSO below the active motor threshold for the corticospinal tract [35]”

I think it would be useful for the authors to mention how many individuals could not withstand the cerebellar stimulation at 70% MSO, and if any individuals displayed evidence of antidromic activation. Moreover I suggest the authors to also add this reference for cerebellar stimulation comfort and reliability: https://doi.org/10.1016/j.brs.2019.09.005

Reviewer #2: General comment

This study aims to elucidate the potential influences of the cerebellum (CB) on neuroplasticity in the primary motor cortex (M1), specifically the role of different indirect wave inputs in M1. To this end, they assessed the magnitude of neuroplasticity of early and late indirect-wave circuits before, during, and after the application of cerebellar tDCS which has been suggested to have modulatory effects on the cerebellar excitability (e.g., reduction of excitability). Based on multiple, complicated though, sets of neurophysiological assessments, the authors concluded that the disruptive effects of CB modulation on M1 plasticity may be selectively mediated by a certain circuit (i.e., late indirect wave circuit).

Although the results are in general complicated, it looks that the authors carefully interpreted each data and explained underlying mechanisms. However, as discussed at the end of the Discussion part, the major issue of this study would be that the sham-controlled design was not adopted for the cerebellar tDCS, leading to the study being less conclusive. In particular, the fact that the cathodal cerebellar tDCS failed to modulate CBI, likely a prerequisite for this study, would have a strong impact enough to ruin all the present findings. Nevertheless, I believe that the study could provide preliminary but suggestive findings in this field to guide future studies in the right direction. Therefore, I think that this study is reasonable to be published in this journal after the authors address the following concerns:

Major comments

1. Although the limitation and the conclusion look carefully reported in the Discussion part, the conclusion in the Abstract reads too strong. The conclusion should be weakened not to mislead the readers' interpretation.

2. The reason why the authors selected downregulating cerebellar stimulation (i.e., cathodal tDCS) is unclear. The authors would better carefully explain the rationale why cathodal but not anodal was selected.

3. How many trials (MEP samples) were finally excluded from the analysis due to voluntary contraction prior to TMS application? The exact number or percentage should be reported in the manuscript.

4. The information of Fig. 2B and C seems redundant, basically the same as Fig. 2A. The authors should better explain why they needed to perform these additional analyzes and prepare these figures. Otherwise, they may better be removed for the sake of simplicity.

7. PLOS authors have the option to publish the peer review history of their article (what does this mean?). If published, this will include your full peer review and any attached files.

Reviewer #1: **Yes: **Danny Spampinato

Reviewer #2: No

---

## [Author Response · Author response to Decision Letter 0]

18 May 2022

Response to Reviewer 1.

1. For the iTMS protocol, can the authors justify why they selected to target the I3-wave (at 4.5 ms) as opposed to (or in addition to) selecting an interval that would predominately target the I2-wave? Since both the I2- and I3-wave get lumped together as “late I-waves”, should we expect a similar finding if one were to target the I2-wave?

Response:

Our decision to target the I3-wave stemmed from existing physiological and functional evidence suggesting that cerebellum may target the circuits responsible for its generation (see paragraph 3 of introduction). In contrast, we are only aware of a single study that has suggested cerebellar interaction with the I2-wave (Ates, Alaydin & Cengiz 2018). Consequently, we felt that focussing on I3 circuits was a more justified approach. While the Ates study suggests that a similar effect on I2 circuits may be possible, the current understanding of these circuits is very limited; indeed, we are unaware of any study that has targeted this interval with iTMS. Consequently, we would prefer to avoid speculating about how these circuits may interact with cerebellum. However, the reviewer makes a good point about the ambiguity with which these two waves are lumped together. We have therefore modified terminology in the revised manuscript (where appropriate) to specifically refer to I3-waves. We have also included a comment about the need for future research to investigate I2-waves more specifically (page 19).

2. The problem with not conducting this additional control condition is that we don’t really know the reliability and reproducibility of iTMS. For instance, what do the authors believe is the error of margin of using the ISI of 4.5 ms for targeting the I3-wave? For instance, Long et al., 2017 (10.1093/brain/awx102) used an ISI of 4.3 ms for their late I-wave protocol, and the authors of this manuscript previously show that an ISI of 4 ms and 5ms yielded greater facilitation on M1 excitability when compared to 4.5ms. Thus, it seems that another potential interpretation of the results could be that the lack of changes on intracortical excitability and corticospinal excitability might be due to the variability of the “late” iTMS protocol and not influences from the cerebellum.

Response:

For the effect of iTMS on PA responses, our previous work (Opie, Cirillo & Semmler 2018; Opie et al. 2021) suggests facilitation in response to a range of ISIs associated with the late I-wave (including 4, 4.1, 4.5, 4.9 and 5 ms) albeit with a reduced response at 4.5ms, as suggested by the reviewer. This response is supported by the facilitation of PA MEPs following iTMS4.5 in the current study. Consequently, we believe that iTMS targeting the late I-wave is able to reliably modulate PA responses. However, the effects of iTMS4.5 on AP responses have not been as well established, and we therefore cannot exclude the possibility that the insensitivity of AP responses stemmed from variable effects of iTMS4.5 on these circuits. This has been acknowledged in the revised manuscript (page 17).

3. Why did the authors select cathodal cerebellar tDCS as their cerebellar neuromodulatory technique as opposed to cerebellar theta burst protocol? Cerebellar tDCS does appear to exclusively produces changes in cerebellar excitability and not corticospinal activity (Galea et al., 2009; 10.1523/JNEUROSCI.2184-09.2009), which has been suggested to produce tonic changes at the Purkinje cell level without activating cerebellar-thal-M1 pathways. However, cerebellar theta burst does yield changes in M1 excitability (10.1016/j.clinph.2008.08.008; 10.1038/srep36191), which might reflect this protocol depolarizing cerebellar Purkinje cells and engaging cerebellar-thal-M1 pathways (that produce changes in M1). Thus, one could presume that cerebellar theta burst would have yielded a more disruptive effect on the iTMS protocol.

Response:

We agree that there is greater evidence supporting the influence of cerebellar TBS on M1 excitability, particularly post-stimulation. However, an ability of cerebellar tDCS to influence M1 activity, both during and after stimulation, has also been shown (Ates, Alaydin & Cengiz 2018; Hamada et al. 2014). Within the current study, we reasoned that coincident application of stimulation over cerebellum and M1 would optimise the chance of observing interactions between these areas. Given that iTMS and tDCS both require ~15 minutes of stimulation, whereas TBS involves 40-190s of stimulation, we therefore decided to apply tDCS. We have clarified this reasoning in the revised manuscript (page 3).

4. While I agree with that different currents are likely recruiting different population of neurons, the previous results by the authors demonstrating changes in AP MEPs following iTMS performed with PA currents (ref 44) does not seem to exactly fit with this notion. One would rather expect that iTMS with PA currents produces greater facilitation of PA MEPs than AP MEPs if distinct populations are being targeted with directional TMS. Thus, if the authors are suggesting that cerebellar tDCS targets AP circuits, would it not be more reasonable to perform iTMS with AP currents?

Response:

Thanks to the reviewer for their comment, we agree that this concept is a problem, and is something we are currently attempting to address in other work. However, this idea is complicated by the higher stimulus intensities required during AP stimulation. This means that it is very unlikely that AP stimulation produces isolated recruitment of AP circuits. This non-specific recruitment is one potential explanation for why we see a potentiation of AP MEPs following PA iTMS (i.e., AP stimulation may be recruiting PA circuits). While the response of AP iTMS to cerebellar modulation would be interesting to investigate, this paradigm has not been characterised previously and will require significant development. It was not feasible to do this within the current study, and we therefore chose to focus on the response to PA iTMS as a preliminary step. Despite this, we have acknowledged this issue in the revised manuscript (page 18).

5. Why did the authors select only 12 trials for each SICF measure, as opposed to 15 trials for each CBI measure and 20 trials for 1 mV assessment? Is SICF considered more reliable than these other measures to justify the much smaller sample?

Response:

The number of trials included in each condition was based on previous literature (Fernandez et al. 2018; Koch et al. 2008; Opie, Cirillo & Semmler 2018). While we are unaware of formal comparisons of reliability between SICF and CBI, our experience with these measurements would suggest greater reliability of SICF. For example, it is unusual for a participant not to show facilitation with SICF, whereas a lack of inhibition following CBI is far more common.

6. I think it would be useful for the authors to mention how many individuals could not withstand the cerebellar stimulation at 70% MSO, and if any individuals displayed evidence of antidromic activation. Moreover I suggest the authors to also add this reference for cerebellar stimulation comfort and reliability: https://doi.org/10.1016/j.brs.2019.09.005

Response:

As suggested, the revised manuscript now lists the number of individuals who could not tolerate CB stimulation at 70%MSO and whether any individuals displayed antidromic activation (page 7).

Thanks to the reviewer for suggesting this reference, it has been included in the revised manuscript.

Response to Reviewer 2.

1. Although the limitation and the conclusion look carefully reported in the Discussion part, the conclusion in the Abstract reads too strong. The conclusion should be weakened not to mislead the readers' interpretation.

Response:

As requested, we have toned down the conclusions included in the abstract.

2. The reason why the authors selected downregulating cerebellar stimulation (i.e., cathodal tDCS) is unclear. The authors would better carefully explain the rationale why cathodal but not anodal was selected.

Response:

One potential outcome from the current study was to inform the design of interventions that could facilitate targeted manipulation of motor functions by selectively modulating specific intracortical circuits of M1. Consequently, we required an intervention that would theoretically potentiate plasticity in M1. As the inhibitory effects of cerebellum on M1 would be expected to reduce the neuroplastic response, we therefore selected an intervention that would reduce cerebellar excitability (i.e., cathodal tDCS). We have clarified this in the revised manuscript (page 3).

3. How many trials (MEP samples) were finally excluded from the analysis due to voluntary contraction prior to TMS application? The exact number or percentage should be reported in the manuscript.

Response:

The percentage of excluded MEP trials is now reported in the revised manuscript (page 9). 

4. The information of Fig. 2B and C seems redundant, basically the same as Fig. 2A. The authors should better explain why they needed to perform these additional analyses and prepare these figures. Otherwise, they may better be removed for the sake of simplicity.

Response:

The reviewer makes a good point. We have removed Fig. 2B and included an explanation for the additional analysis (page 12) for Fig. 2C (now relabelled Fig. 2B).

---

## [Decision Letter · Decision Letter 1]

22 Jun 2022

PONE-D-22-03125R1Cerebellar transcranial direct current stimulation disrupts neuroplasticity of intracortical motor circuits.PLOS ONE

Dear Dr. Opie,

Thank you for submitting your manuscript to PLOS ONE. After careful consideration, we feel that it has merit but does not fully meet PLOS ONE’s publication criteria as it currently stands. Therefore, we invite you to submit a revised version of the manuscript that addresses the points raised during the review process.

The Reviewers were mostly satisfied with the revised version. Your manuscript is basically accepted. Reviewer #2 has only two minor issues for you to address. I am sure you can proceed quickly and resubmit a revised version.

We look forward to receiving your revised manuscript.

Kind regards,

François Tremblay, PhD

Academic Editor

PLOS ONE

Journal Requirements:

Reviewers' comments:

Reviewer's Responses to Questions

**Comments to the Author**

1. Does the manuscript adhere to the experimental procedures and analyses described in the Registered Report Protocol?

If the manuscript reports any deviations from the planned experimental procedures and analyses, those must be reasonable and adequately justified.

Reviewer #1: Yes

Reviewer #2: Yes

2. If the manuscript reports exploratory analyses or experimental procedures not outlined in the original Registered Report Protocol, are these reasonable, justified and methodologically sound?

A Registered Report may include valid exploratory analyses not previously outlined in the Registered Report Protocol, as long as they are described as such.

Reviewer #1: Yes

Reviewer #2: Yes

3. Are the conclusions supported by the data and do they address the research question presented in the Registered Report Protocol?

The manuscript must describe a technically sound piece of scientific research with data that supports the conclusions. The conclusions must be drawn appropriately based on the research question(s) outlined in the Registered Report Protocol and on the data presented.

Reviewer #1: Yes

Reviewer #2: Yes

4. Have the authors made all data underlying the findings in their manuscript fully available?

Reviewer #1: Yes

Reviewer #2: Yes

5. Is the manuscript presented in an intelligible fashion and written in standard English?

Reviewer #1: Yes

Reviewer #2: Yes

6. Review Comments to the Author

Please use the space provided to explain your answers to the questions above. (Please upload your review as an attachment if it exceeds 20,000 characters)

Reviewer #1: The authors have addressed my concerns.

My final suggestion would be for the authors to include two recent papers in the introduction paragraph #3, which have shown the interaction of cerebellar projections to specific excitatory (DOI: 10.1523/JNEUROSCI.2397-19.2020) and inhibitory (DOI: 10.1016/j.brs.2021.08.022) interneurons.

Reviewer #2: General comment

Although the authors addressed my previous comments and fairly modified the manuscript, there remain some points that need to be further or properly clarified before getting accepted. I hope that the authors address the following comments accordingly.

Specific comments

#1. The rationale for selecting cathodal but not anodal remains unclear. The authors explained in the revised manuscript that “previous work has reported that tDCS to CB modulates concurrent plasticity induction of M1 [15: Hamada et al., 2014]” followed by “therefore, we investigated whether enhancement of M1 plasticity could be achieved by downregulating CB excitability using cathodal tDCS~”. Hamada et al. (2014) demonstrated that concurrent anodal cerebellar tDCS abolishes M1 plasticity, without testing with cathodal one. Hamada et al. (Journal of Physiology, 2012), on which Hamada et al. (2014) rely, also found that both anodal and cathodal cerebellar tDCS block M1 plasticity. These previous findings do not look directly support the authors’ decision to select cathodal but not anodal. A more careful and logical explanation should be achieved.

#2. Is 1.9% the average of excluded MEPs among participants? Or, is it the percentage of pooled MEPs among all participants and conditions, etc.? Please specify in more detail.

7. PLOS authors have the option to publish the peer review history of their article (what does this mean?). If published, this will include your full peer review and any attached files.

Reviewer #1: **Yes: **Danny Spampinato

Reviewer #2: No

---

## [Author Response · Author response to Decision Letter 1]

23 Jun 2022

Response to Reviewer 1

1. My final suggestion would be for the authors to include two recent papers in the introduction paragraph #3, which have shown the interaction of cerebellar projections to specific excitatory (DOI: 10.1523/JNEUROSCI.2397-19.2020) and inhibitory (DOI: 10.1016/j.brs.2021.08.022) interneurons.

Response:

The reviewers J Neurosci article has now been referenced in paragraph 3 of the introduction. Regarding the Brain Stimul article, while we agree it supports the concept that cerebellum has specific interactions with different intracortical circuits in M1, the implications with respect to early vs late I-waves remain less clear. Given that the point of this paragraph is to describe the evidence supporting differential effects of cerebellum on early and late I-wave circuits, we would therefore prefer to not include reference to this article in this paragraph.

Response to Reviewer 2

1. The rationale for selecting cathodal but not anodal remains unclear. The authors explained in the revised manuscript that “previous work has reported that tDCS to CB modulates concurrent plasticity induction of M1 [15: Hamada et al., 2014]” followed by “therefore, we investigated whether enhancement of M1 plasticity could be achieved by downregulating CB excitability using cathodal tDCS~”. Hamada et al. (2014) demonstrated that concurrent anodal cerebellar tDCS abolishes M1 plasticity, without testing with cathodal one. Hamada et al. (Journal of Physiology, 2012), on which Hamada et al. (2014) rely, also found that both anodal and cathodal cerebellar tDCS block M1 plasticity. These previous findings do not look directly support the authors’ decision to select cathodal but not anodal. A more careful and logical explanation should be achieved.

Response

The outcomes reported by Hamada and colleagues related to the response to paired-associative stimulation (PAS), and were suggested to be mediated by tDCS (both cathodal and anodal) over cerebellum interrupting afferent input that underpins the spike-timing dependent response to PAS. Consequently, these responses were not thought to be driven by changes to how cerebellum influences motor cortex (M1) directly, but by tDCS over cerebellum influencing afferent input to M1. In contrast, the iTMS paradigm we applied is not dependent on afferent input that might be influenced by cerebellar tDCS, and this was not how we were trying to influence M1 plasticity (we discuss this point on page 15 of the manuscript). Subsequently, we did not expect that cerebellar tDCS within our study would influence M1 plasticity in the same way as was reported by the Hamada studies. Instead, the goal of our intervention was to modulate the direct influence of cerebellum on M1. Specifically, cerebello-thalamo-cortical projections to M1 have an inhibitory effect, and it was therefore possible that reducing this inhibitory tone could produce an environment within M1 that was more amenable to neuroplasticity induction. As previous work has shown that cathodal tDCS over cerebellum is able to achieve this (i.e., CBI is reduced by cerebellar cathodal tDCS; Galea et al., 2009), we decide to apply cathodal stimulation. We have clarified this rationale within the revised manuscript (page 3/4). Furthermore, as we understand why reference to the Hamada papers would have appeared counter intuitive, reference to these papers has been removed from paragraph 2 of the introduction.

2. Is 1.9% the average of excluded MEPs among participants? Or, is it the percentage of pooled MEPs among all participants and conditions, etc.? Please specify in more detail.

Response:

This represents the percentage of MEPs excluded across all participants. This has been clarified within the revised manuscript.

---

## [Editor Report · Decision Letter 2]

29 Jun 2022

Cerebellar transcranial direct current stimulation disrupts neuroplasticity of intracortical motor circuits.

PONE-D-22-03125R2

Dear Dr. Opie,

We’re pleased to inform you that your manuscript has been judged scientifically suitable for publication and will be formally accepted for publication once it meets all outstanding technical requirements.

Kind regards,

François Tremblay, PhD

Academic Editor

PLOS ONE
---

## [Editor Report · Acceptance letter]

4 Jul 2022

PONE-D-22-03125R2 

Cerebellar transcranial direct current stimulation disrupts neuroplasticity of intracortical motor circuits. 

Dear Dr. Opie:

I'm pleased to inform you that your manuscript has been deemed suitable for publication in PLOS ONE. Congratulations! Your manuscript is now with our production department. 

Kind regards, 

on behalf of

Dr. François Tremblay 

Academic Editor

PLOS ONE